# CounteRGAN: Generating Counterfactuals for Real-Time Recourse and Interpretability using Residual GANs

**Daniel Nemirovsky**[*1]     **Nicolas Thiebaut**[2]     **Ye Xu**[*3]     **Abhishek Gupta**[*3]

[1] Amazon, Seattle, WA, U.S.A., nemird@amazon.com
[2] Hired, New York, NY, U.S.A., nicolas.thiebaut@hired.com
[3] Meta, Menlo Park, CA, U.S.A., {yexu, abigupta}@fb.com

## Abstract

Model interpretability, fairness, and recourse for end users have increased as machine learning models have become increasingly popular in areas including criminal justice, finance, healthcare, and job marketplaces. This work presents a novel method of addressing these issues by producing *meaningful counterfactuals* that are aimed at providing recourse to users and highlighting potential model biases. A meaningful counterfactual is a reasonable alternative scenario that illustrates how input data perturbations can influence the model's output. The *CounteRGAN* method generates meaningful counterfactuals for a target classifier by utilizing a novel *Residual Generative Adversarial Network (RGAN)*. We compare our method against leading state-of-the-art approaches on image and tabular datasets over a variety of performance metrics. The results indicate a significant improvement over existing techniques in combined metric performance, with a latency reduction of 2 to 7 orders of magnitude which enables providing real-time recourse to users. The code for reproducibility can be found here: https://github.com/gan-counterfactuals/countergan.

## 1 INTRODUCTION

A growing number of domains use machine learning (ML) predictive models on a daily basis, such as criminal justice for predicting recidivism [Tollenaar and van der Heijden, 2013], healthcare for diagnosing patients [Miotto et al., 2018], job marketplaces for hiring candidates [Faliagka et al., 2012], and finance for loan approvals [Addo et al., 2018]. The pervasiveness of this technology has resulted

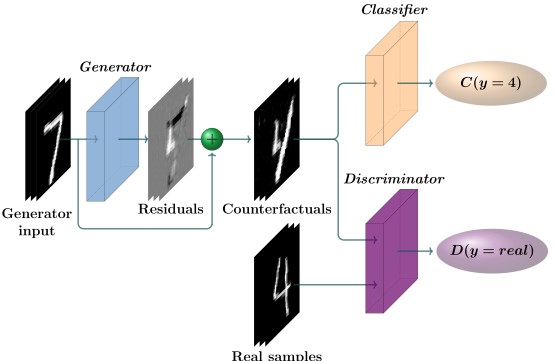

Figure 1: The CounteRGAN method applied to an example from MNIST. Three neural nets are employed: a residual generator, a discriminator that distinguishes the real data, and a target classifier. The loss function of the generator uses both the classifier and discriminator output. In the example, residuals are being produced by the generator, which, when added to the input, creates realistic images of a "4".

in a growing need for model interpretability as well as discussions regarding "the right to explanation" in the legal and machine learning communities [Wachter et al., 2017a, Selbst and Powles, 2017, Goodman and Flaxman, 2016]. Consequently, considerable resources have been allocated not only to improve prediction explainability, but also to provide recourse to users to enhance fairness and opportunity.

A number of leading explainability methods [Ribeiro et al., 2016, Lundberg and Lee, 2017, Sundararajan et al., 2017, Selvaraju et al., 2019, Chattopadhay et al., 2018] have shown promise in shedding light on the opaque logic and feature influences behind a prediction model. By answering *why* a model predicted the result it did, explainability methods are useful for validating training and identifying possible racial, social, and gender biases. Alternatively, recourse aims to provide the user with actionable feedback by showing *how* a prediction can be altered or reversed. By recommending

---

[*]Work done while at Hired.

*Accepted for the 38th Conference on Uncertainty in Artificial Intelligence* (UAI 2022).

certain changes to the input data, recourse can inform a user how to improve their chances of receiving a better diagnosis, getting a loan, or getting hired.

Alternative hypothetical scenarios that rely on perturbations to the original input values are known as *counterfactuals*. The impacts on a prediction from the changes suggested by counterfactuals can be useful for model interpretability as well as for providing recourse to users. If, for instance, one's gender or race are suggested to be changed to alter the prediction result, then predictor biases could be suspected. In contrast, recourse relies on providing interpretable feedback users can act upon and which helps them change the prediction result in their favor. Counterfactuals should be meaningful in order to provide recourse. *Meaningful counterfactuals* must be *realistic*, *computationally efficient*, and able to provide *actionable* feedback to the user that would help them achieve the *desired prediction* outcome.

Unlike computing efficiency and prediction outcomes which can be quantified by examining latency and predictor score, realism and actionability are fundamentally more subjective. A counterfactual is said to be realistic when it closely resembles or "fits in" with the known distribution of data. For example, a house with a negative number of bedrooms is evidently unrealistic. A less obvious example, however, would be of a house with a seemingly extreme layout but where realism is dependent upon the location and society.[1] By contrast, actionability pertains to whether proposed changes are interpretable and reasonable for a user to take action. For instance, increasing one's body mass index, learning a new programming language, or reducing outstanding debt are all actionable changes (although some are more difficult than others). The proximity and sparsity of a counterfactual can serve as an intuitive but imperfect proxy for actionability since they indicate the magnitude and number of potential perturbations in the counterfactual. A realistic counterfactual doesn't necessarily result in actionable change. As an example, it is not reasonable to reduce one's age or education even if it results in a counterfactual which is very similar to a realistic individual. Moreover, depending on the use case, actionable changes may result in unrealistic counterfactuals. For instance, manipulating pixels and text or fixing features to specific values can confuse the target classifier in a manner similar to adversarial attacks which exploit seldom used regions of a classifier's decision boundary.

Existing recourse methods [Wachter et al., 2017b, Van Looveren and Klaise, 2019, Mothilal et al., 2020] employ variations of regularized gradient descent to perform the counterfactual search. As a result, these are severely constrained by latency since a counterfactual search is required for every

unique input data point. A lack of counterfactual realism also affects algorithms that do not explicitly consider realism constraints [Wachter et al., 2017b] or conflate realism with actionability [Mothilal et al., 2020]. Latency constraints and the distinction between realism and actionability are key to framing the counterfactual search problem as a natural fit for Generative Adversarial Networks (GANs). GANs [Goodfellow et al., 2014a] are a class of ML models capable of producing strikingly realistic synthetic data with low and fixed latencies. These models formulate the training of two artificial neural networks, a generator and a discriminator, as an adversarial game. While the discriminator is trained to recognize realistic data, the generator is trained to synthesize data that can fool the discriminator. Effectively trained generators produce realistic data requiring a single forward-pass through the neural network.

In this work, we formalize a *Residual GAN (RGAN)* architecture, useful for generating perturbations directly and alleviating mode collapse. The latter issue can occur during training when the generator model begins to consistently produce similar or identical outputs regardless of the inputs. The RGAN is used in conjunction with a fixed target classifier to generate meaningful counterfactuals that are suitable for providing recourse to users and improving model interpretability and fairness. This new approach, referred to as CounteRGAN, produces counterfactuals that meet or exceed the predictive gain and actionability of two state-of-the-art methods while improving realism and decreasing latency by 2-7 orders of magnitude. Figure 1 provides a clarifying illustration of the CounteRGAN architecture applied to an example from MNIST. The proposed technique can be used to provide real-time recourse to users of ML predictors employed in a wide range of industries. Our goal is to facilitate improved opportunities, transparency, and fairness associated with ML prediction. The main contributions of this work include:

- The application of GANs to produce meaningful counterfactuals that can provide real-time recourse to users as well as improved model interpretability and fairness.

- Formalizing a novel *Residual Generative Adversarial Network (RGAN)* that trains the generator to produce residuals that are intuitive to the notion of perturbations used in counterfactual search. This model is also shown to alleviate mode collapse.

- The *CounteRGAN* method which applies an RGAN model in conjunction with a target classifier to produce meaningful counterfactuals 2 to 7 orders of magnitude faster than existing methods. A second variant is also introduced for when the target classifier's gradients or architecture is unknown (e.g., a black-box model).

---

[1] One of the authors recalls the wonderment of seeing the tall and narrow Dutch houses neatly packed into picturesque rows lining idyllic canals. Consider how surreal such homes would appear in the Andean mountain villages or vice versa.

## 2 RELATED WORK

Influential and relevant previous work comes from both the counterfactual and GAN domains.

### 2.1 COUNTERFACTUALS

Borrowing from philosophy and causality [Lewis, 1973, Pearl, 2009, Karimi et al., 2020], counterfactuals were introduced as explanations for ML predictors by Wachter et al. [Wachter et al., 2017b]. The authors formulated counterfactual search as a minimization problem with an added regularization term to enforce feature perturbation sparsity. Given an original data point $x$ and a ML classifier $C$, the counterfactual $x_{\text{cf}}$ is produced using iterations of gradient descent to increase the classifier's prediction $C_t\left(x_{\text{cf}}\right)$ for a given target class $t$. This approach is useful for producing counterfactuals of the desired class but tends to be slow and results may be unrealistic.

Several approaches have targeted increasing counterfactual realism. These include a graph-based density approach [Poyiadzi et al., 2019] and applying an autoencoder reconstruction error term to constrict the counterfactual from straying too far from the observed feature space [Dhurandhar et al., 2018, Joshi et al., 2019, Pawelczyk et al., 2020]. An alternative approach [Mothilal et al., 2020] focuses on producing multiple diverse counterfactuals for each query instance such that the user can select the most relevant. A novel technique proposed utilizing class prototypes [Kim et al., 2016] to guide the counterfactual search toward high-density regions of the feature space [Van Looveren and Klaise, 2019]. While the aforementioned methods are limited to differentiable classifiers, a heuristic search involving "growing spheres" is used [Laugel et al., 2017] to produce sparse counterfactuals for non-differentiable or black-box models. This method, however, does not further address realism nor latency concerns. All of the approaches mentioned above suffer from high computational latencies. The proposed CounteRGAN method, however, is able to produce meaningful counterfactuals within real-time latency constraints for both differentiable and non-differentiable models.

Counterfactuals are also produced in adversarial perturbation techniques [Goodfellow et al., 2014b]. For example, modifying a single pixel in an image of a horse to fool a classifier into predicting it is an image of a frog [Su et al., 2017]. In general, these methods are aimed at confusing a target classifier without necessarily providing meaningful recourse to users; a task that requires balancing desired prediction with realism and actionability.

### 2.2 GENERATIVE ADVERSARIAL NETS (GANS)

The introduction of GANs [Goodfellow et al., 2014a] marked a milestone in the field of generative models. The elegance of a GAN lies in its formulation of training as an adversarial minimax game between two differentiable models able to approximate probability distributions utilizing backpropagation and gradient descent. Interest in GANs has since intensified and several novel approaches have been proposed towards improving training [Salimans et al., 2016, Arjovsky et al., 2017] and architecture [Radford et al., 2015, Denton et al., 2015, Zhang et al., 2016]. Providing additional input such as label information to condition GANs, for example, to generate specific MNIST digits, has been previously proposed [Mirza and Osindero, 2014, Odena et al., 2017]. GANs have also been applied to problems that share intuitive notions with counterfactuals such as representation learning [Chen et al., 2016, Tran et al., 2017], image-to-image translation [Isola et al., 2016, Zhu et al., 2017a,b], style transfer [Huang and Belongie, 2017, Karras et al., 2020], and illumination [Wang et al., 2017, Zhang et al., 2019]. The use of GANs with residual images has been proposed for attribute manipulation in images [Shen and Liu, 2016]. These methods are domain-specific and often target realism instead of reverting decisions of existing classifiers and providing actionable feedback to users. An unrelated but similarly termed "Residual GAN" [Tavakolian et al., 2019] uses a deep residual convolutional network to a generator to magnify subtle facial variations. In contrast, we define and use a Residual GAN, where the generator is trained to synthesize residuals directly. Unlike prior work, and to the best of our knowledge, we are the first to apply GANs towards the generation of meaningful counterfactuals for recourse.

## 3 GAN-BASED COUNTERFACTUAL GENERATION

To overcome the mode collapse and actionability limitations of applying standard GANs to counterfactual generation, we formalize the Residual GAN (RGAN) as a special case of GAN. The CounteRGAN, by contrast, is the proposed technique that couples an RGAN with a target classifier to synthesize meaningful counterfactuals.

### 3.1 RESIDUAL GAN (RGAN)

Similar to how conditional GANs [Mirza and Osindero, 2014], though initially motivated by image synthesis, have been generalized to be applicable to several domains, we also introduce a generalized RGAN formulation, whose original motivation stemmed from generating counterfactuals, but could also be applied to other domains including image synthesis and photo editing [Zhang et al., 2019]. The

generalized RGAN is a special instance of a GAN where the generator generates residuals rather than a complete synthetic data point. As in standard GANs, a discriminator $D$ and a generator $G$ are trained in a minimax game framework where the generator seeks to minimize and the discriminator aims to maximize the following value function:

$$
\begin{aligned}
\mathcal{V}_{\mathrm{RGAN}}(D,G) =& \mathbb{E}_{x \sim p_{\mathrm{data}}} \log D(x) \\
&+ \mathbb{E}_{z \sim p_{\mathrm{z}}} \log \left(1 - D(z + G(z))\right),
\end{aligned} \quad (1)
$$

where the generator's input $z \in \mathbb{Z}$ is a latent variable sampled from a probability distribution $p_{\mathrm{z}}$. The input to the RGAN discriminator is $z + G(z)$, as opposed to the standard GAN which utilizes $G(z)$ directly.

The generalized RGAN formulation restricts the dimensionality of the latent (input) space to be the same as the data feature (output) space ($\mathbb{Z} = \mathbb{X}$).[2] and forces the generator to learn contingent relationships between its input and output. This constraint enables fine-grained regularization directly on the residuals [3] and helps to alleviate mode collapse caused when the GAN generates similar output regardless of its input which it learns to ignore.

## 3.2 COUNTERGAN

The proposed counterfactual search method, termed CounteRGAN, utilizes an RGAN and a fixed target classifier $C$ to produce meaningful counterfactuals for providing recourse to users and improved interpretability. The method is capable of producing counterfactuals that are of the desired target class, realistic, actionable, and require low computational latency. Below we present two variants of the CounteRGAN value function for when the classifier's gradients are and are not known. The search process seeks to maximize the value function with respect to the discriminator $D$ and minimize it with respect to the generator $G$.

If the classifier is known and differentiable, then the following CounteRGAN value function can be used:

$$
\begin{aligned}
\mathcal{V}_{\mathrm{CounterRGAN}}(G,D) =& \mathcal{V}_{\mathrm{RGAN}}(G,D) + \mathcal{V}_{\mathrm{CF}}(G,C,t) \\
&+ \mathrm{Reg}(G(x)), \quad (2)
\end{aligned}
$$

where $t$ is the target class. The first term ($\mathcal{V}_{\mathrm{RGAN}}$) uses a

---

[2]This constraint could be overcome by utilizing an autoencoder. The synthesized data point $z + G(x) \in \mathbb{Z}$ can then be decoded to a new data point in the same space as the input data, such that $\mathrm{decoder}(z + G(x)) \in \mathbb{X}$

[3]Note that the activation function for the generator's output layer constrains the residuals and therefore their impact on the final synthesized output. Thus, depending on the scenario, it is recommended to use a symmetric activation function (e.g., linear, tanh) capable of outputting positive and negative values within the same order of magnitude as the input features.

specialized RGAN that reads:

$$
\begin{aligned}
\mathcal{V}_{\mathrm{RGAN}}(D,G) =& \mathbb{E}_{x \sim p_{\mathrm{data}}} \log D(x) \\
&+ \mathbb{E}_{x \sim p_{\mathrm{data}}} \log \left(1 - D(x + G(x))\right),
\end{aligned} \quad (3)
$$

where both the generator $G$ and discriminator $D$ use inputs samples $x$ from the same probability distribution $p_{\mathrm{data}}$. In isolation, this formulation would result in the generator learning to systematically output null residuals since the inputs are already realistic data. However, since the generator is also required to account for the classifier's loss term $\mathcal{V}_{\mathrm{CF}}$, this formulation helps to enforce counterfactual realism.

The term ($\mathcal{V}_{\mathrm{CF}}$) drives the counterfactual toward the desired class $t$, it reads:

$$
\mathcal{V}_{\mathrm{CF}}(G,C,y) = \mathbb{E}_{x \sim p_{\mathrm{data}}} \log \left(1 - C_t(x + G(x))\right), \quad (4)
$$

where $C_t$ is the classifier's prediction function for the desired class.

The last term of the CounteRGAN value function, $\mathrm{Reg}(G(x))$, can be any weighted combination of L1 and L2 regularization terms and helps to control the sparsity and amplitude of the residuals (i.e., feature perturbations) which serves as a proxy for counterfactual actionability.

While most existing counterfactual search methods target differentiable models, the target classifiers used in production settings may often be non-differentiable or unknown (black-box).[4] To account for such scenarios, we introduce a second CounteRGAN value function termed CounteRGAN-bb for black-box models. Instead of computing a classifier's gradients, this variant weighs the first term of the RGAN value function by the classifier's prediction score $C_t(x_i)$ such that the corresponding value function reads

$$
\begin{aligned}
\mathcal{V}_{\mathrm{CounteRGAN-bb}}(D,G) =& \frac{\sum_i C_t(x_i) \log D(x_i)}{\sum_i C_t(x_i)} \\
&+ \frac{1}{N} \sum_i \log \left(1 - D(x_i + G(x_i))\right) + \mathrm{Reg}(G, \{x_i\}),
\end{aligned}
$$

$$(5)$$

where $\mathrm{Reg}(G, \{x_i\})$ is analogous to the regularization term introduced previously and samples $x_i$ are drawn from the entire data distribution.

$$
\mathrm{Reg}\left(G, \{x_i\}\right) = \alpha \sum_i \|G(x_i)\|_1 + \beta \sum_i \|G(x_i)\|_2^2.
$$

The specific form of this value function is motivated by the resulting convergence properties, formalized by Theorem 1 below. The proof of convergence is provided in the supplementary material.

---

[4]For example, while a bank employee may have access the a loan classifier's architecture, the same cannot necessarily be said about the customer or a third-party service.

**Theorem 1** *If the discriminator is systematically allowed to reach its optimum, and the generator has sufficient capacity, then the minimax optimization of the value function from equation 5 converges to the Nash equilibrium. The full generator's output distribution $p_{g_+}$ converges to a distribution $p_{C_t}$ defined by*

$$p_{C_t}(x) = \mathcal{N}_t \, C_t(x) \, p_{\text{data}}(x), \qquad (6)$$

*where $N_t$ is a normalization constant.*[5]

Using either value function variant, the CounteRGAN discriminator learns to discriminate between real and synthetic data points, while the generator aims to balance the desired classification with realism and sparsity (actionability) constraints. As a result, the generator learns to produce residuals that, when added to the input, produce realistic and sparse counterfactuals that are classified by $C$ to be as close to 1 for the desired class as possible. Once trained, the generator can produce counterfactuals quickly via a single forward pass through the neural network.

## 4 EXPERIMENTS

We compare the proposed CounteRGAN approach against two state-of-the-art counterfactual search methods [Wachter et al., 2017b, Van Looveren and Klaise, 2019]. As far as possible, our experiments mirror the experimental setups used in those proposals including the evaluation datasets and model architectures. With the exception of [Wachter et al., 2017b] which is a foundational work and conventional baseline, other counterfactual search methods mentioned in the Related Work are not included either because they do not address realism [Mothilal et al., 2020, Laugel et al., 2017] or because their latency is prohibitive for real-time applications [Poyiadzi et al., 2019]. The first experiment is conducted using the MNIST handwritten digit dataset [LeCun and Cortes, 2010] which lends to providing visual clarity of each method's approach. The second experiment makes use of the COMPAS recidivism dataset [ProPublica, 2017] to highlight how meaningful counterfactuals can be helpful for improving model interpretability and fairness. We added a third experiment in the supplementary material, which uses an Indian diabetes dataset [Smith et al., 1988] and helps to demonstrate that the CounteRGAN is also effective on tabular data when some of the features are immutable.

**Methods** Given an input data point $x_i$, all methods described below aim to produce a counterfactual $x_i^{\text{cf}}$ that a target classifier $C$ will predict as the desired class.

| Metric | Formula |
|---|---|
| Counterfactual prediction gain | $\mathbb{E}\left[C(x_i^{\text{cf}}) - C(x_i)\right]$ |
| Realism | $\mathbb{E}\left[\left\|\text{AE}\left(x_i^{\text{cf}}\right) - x_i^{\text{cf}}\right\|_2^2\right]$ |
| Actionability (Sparsity & proximity) | $\mathbb{E}\left[\left\|x_i^{\text{cf}} - x_i\right\|_1\right]$ |
| Latency | $\mathbb{E}\left[\delta t_i\right]$ |

Table 1: Evaluation metrics summary. $C$ is the target classifier and $x_i$ denotes the data point for which a counterfactual ($x_i^{\text{cf}}$) is sought. An autoencoder (AE) is used to reconstruct $x_i^{\text{cf}}$. Expectations are computed over the test sets.

- *Regularized Gradient Descent (RGD)*: a gradient descent based counterfactual search [Wachter et al., 2017b] that minimizes the sums of the squared differences between the desired outcome and the counterfactual. A regularization term is used to enforce sparsity.[6]

- *Counterfactual Search Guided by Prototypes (CSGP)*: this method [Van Looveren and Klaise, 2019] extends RGD by using class prototypes to push the counterfactual towards a more realistic data point of the desired class. The value function is modified to include a distance measure from the counterfactual to the class prototype in latent space ($L_{\text{proto}}$).

- *Standard GAN (GAN)*: This method applies a standard GAN [Goodfellow et al., 2014a], in conjunction with the target classifier $C$. The generator is modified to use real data points as input (as opposed to random latent variables) and synthesize complete counterfactuals.

- *CounteRGAN*: The proposed method from section 3 that uses the specialized RGAN together with the target classifier $C$. The value function from Equation 2 is used when $C$ is a white-box model (i.e., known gradients) and Equation 5 is used when $C$ is a black-box model (i.e., unknown or undefined gradients).

**Evaluation metrics** To evaluate the relative performance of the methods, we identify four desirable properties of counterfactual generation and propose the corresponding metrics detailed below (see Table 1 for a summary). These metrics are based on those found in related work and we have also added established measures of realism and actionability. All metric results from the experiments, except for batch latency, are based on averages of individually computed counterfactuals using the test data. Batch latency is the total computation time necessary to produce counterfactuals for an entire batch. Each table presents the results of the

---

[5]Explicitly, $\mathcal{N}_t = \left(\int C_t(x) \, p_{\text{data}}(x)\mathrm{d}x\right)^{-1}$ but it doesn't need to be computed for our purpose.

[6]For this method and the next, we use the implementations (including gradient approximating versions for black-box models) provided by https://github.com/SeldonIO/alibi.

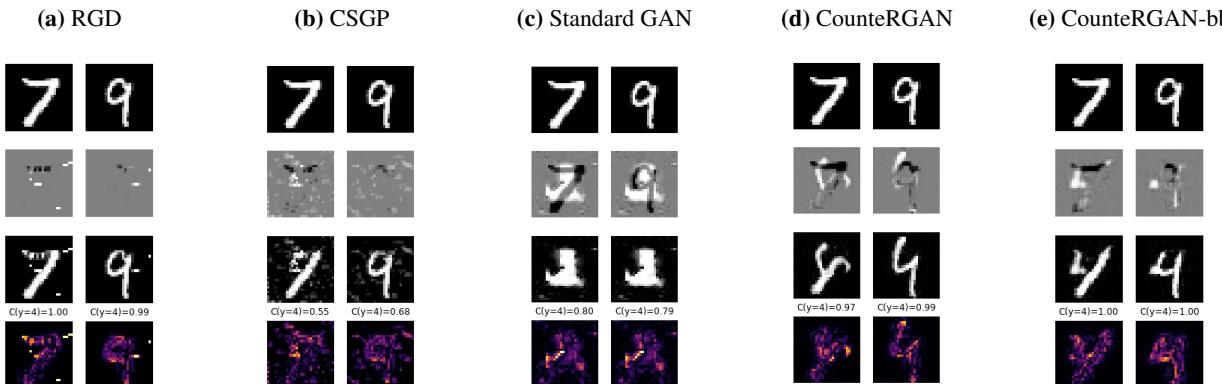

| **(a)** RGD | **(b)** CSGP | **(c)** Standard GAN | **(d)** CounteRGAN | **(e)** CounteRGAN-bb |

Figure 2: Comparison of counterfactual examples produced by different methods on MNIST. Given two separate digit images (7 and 9), each method is tasked with producing counterfactuals that the classifier will predict as a "4". The first row shows the original input image. The second row highlights the perturbations that the counterfactual produces (residuals in the case of CounteRGAN). Negative perturbation values are black, positive values are white, and null or zero values are grey. The third row shows the final counterfactual produced after adding the input with the perturbations. The fourth and final row displays the autoencoder reconstruction error with brighter points representing less realism. Existing methods (a) and (b) result in less realistic counterfactuals. Method (c) lacks realism as well as actionability due to mode collapse. The CounteRGAN methods (d) and (e) (black-box) result in the most realistic counterfactuals.

|  | White-box classifier | | | | Black-box classifier | | |
|---|---|---|---|---|---|---|---|
|  | RGD | CSGP | GAN | CounteRGAN | RGD | CSGP | CounteRGAN |
| ↑ Prediction gain | **0.83 ± 0.01** | 0.43 ± 0.00 | 0.69 ± 0.01 | 0.80 ± 0.01 | 0.45 ± 0.01 | 0.41 ± 0.00 | **0.85 ± 0.01** |
| ↓ Realism | 4.56 ± 0.01 | 4.58 ± 0.01 | 4.50 ± 0.00 | **3.95 ± 0.01** | 3.94 ± 0.01 | **3.58 ± 0.01** | 4.37 ± 0.01 |
| ↓ Actionability | **20.63 ± 0.41** | 54.24 ± 0.60 | 151.98 ± 0.43 | 79.47 ± 0.47 | **31.86 ± 0.61** | 48.79 ± 1.69 | 72.99 ± 0.52 |
| ↓ Latency (ms) | 4,129.57 ± 3.33 | 5,359.58 ± 2.72 | **13.05 ± 0.04** | 13.33 ± 0.04 | 8,464.10 ± 42.54 | 30,235.47 ± 553.47 | **13.52 ± 0.04** |
| ↓ Batch latency (s) | 4,129,570 | 5,359,580 | **45** | 45 | 84,641,012 | 302,354,681 | **45** |

Table 2: MNIST test data results (mean and 95% confidence interval). The arrows indicate whether larger ↑ or lower ↓ values are better, and the best results are in bold. The realism metric typically ranges from 3.89 (mean reconstruction error on the test set) to 11.99 (reconstruction error random uniform noise $[0, 1]$). Computations are performed using the entire test set (10,000 samples).

methods assuming that the target classifier's gradients are known (white-box model) or unknown (black-box model).

- *Prediction gain*: the difference between the classifier's prediction on the counterfactual ($C_t(x_i^{\text{cf}})$) and the input data point ($C_t(x_i)$), for the target class $t$. Since the maximum score classifier $C$ can predict is 1, the range for prediction gain is $[0, 1]$ with higher gain indicating more improvement.

- *Realism*: a measure of how well a counterfactual "fits in" with the known data distribution. We adopt a strategy inspired by [Van Looveren and Klaise, 2019, Dhurandhar et al., 2018], in which we train a denoising autoencoder AE $(\cdot)$ on the training set and use the L2 norm of the reconstruction error as a measure of realism. A lower value represents higher realism.

- *Actionability (sparsity & proximity)*: Sparsity and proximity are commonly used [Mothilal et al., 2020],

though imperfect, proxies for true actionability which is inherently difficult to quantify. We follow existing methods which have measured actionability as a function of the number and magnitude of perturbations present in the counterfactual ($x_i^{\text{cf}}$) relative to the input data point ($x_i$) using the L1 norm. Lower values indicate fewer changes and therefore a higher degree of actionable feedback. Weighting the sparsity penalty according to the degree of feature mutability could be promising for future work. In the supplemental materials, an experiment using the Pima Indian Diabetes dataset [Smith et al., 1988] is provided as an example.

- *Latency*: the computational latency needed to generate counterfactuals. Individual counterfactual computations can impact real-time applicability. Batch results are useful to highlight scalability limitations since large amounts of counterfactuals may be desired to be generated without real-time constraints but within practical

latency and cost budgets. Lower values are better and subsecond latencies are necessary for real-time applicability.

## 4.1 EXPERIMENT USING MNIST IMAGE DATASET

MNIST consists of 70,000 images of handwritten digits (28x28 black and white pixels, that we normalize to have values between 0 and 1) with equal amounts of samples for each digit class. The images are split for training and testing with 60,000 and 10,000 samples respectively, both of which are balanced in terms of labels.

A convolutional neural network (CNN) is used as the target classifier which is trained to correctly classify the digits (98.6% accuracy on the test set). In addition to the classifier, we train a denoising convolutional autoencoder that is used to gauge counterfactual realism. Each method is tasked with generating counterfactuals that the classifier should predict as a "4" digit. All results are based on the averages from generating counterfactuals for all of the 10,000 samples from the test set.

Examples of counterfactuals for two digits are shown in Figure 2. All methods succeed in producing counterfactuals that the classifier labels as "4", with predicted probabilities ranging from 0.55 to 1. In RGD (Figure 2a), counterfactuals resemble adversarial attacks in that they are composed of subtle perturbations that lead to the desired classification, but are highly unrealistic. The CSGP algorithm (Figure 2b) seems to perform better visually, affecting relevant pixels to turn the digits into the desired "4" but still lacks realism. The counterfactual search with a regular GAN (Figure 2c) saliently exhibits mode collapse. Without the residual formulation, the generator simply learns to generate the same image regardless of the input. The two CounteRGAN formulations (Figures 2d and 2e) output visually convincing counterfactuals, as corroborated by the large classifier scores (0.97 to 1) and low autoencoder reconstruction errors.

The complete metrics results for the MNIST dataset are presented in Table 2. While all methods largely increase the prediction of the target class, CSGP is noticeably less impactful. The RGD method outputs sparser counterfactuals at the significant cost of realism. The two CounteRGAN variants, by contrast, generate the most realistic counterfactuals with high actionability and prediction gain. Notably, the GAN and proposed CounteRGAN approaches also achieve >300x and >600x latency improvements over existing methods when generating single counterfactuals on white-box and black-box classifiers respectively. On a batch of the full 10000 samples from the test set, the GAN based methods achieve an impressive 5 to 7 orders of magnitude improvement.

## 4.2 EXPERIMENT USING COMPAS RECIDIVISM DATASET

Predictive models can have life-changing effects on individuals in certain situations. In the United States, for example, recidivism prediction models such as the COMPAS score [ProPublica, 2017] are used to guide sentencing for crimes in several states and major cities. As this experiment demonstrates, meaningful counterfactuals can be used to improve model interpretability and fairness by exposing harmful biases such as racial and gender biases.

The COMPAS dataset consists of 7,214 arrests logged in Broward County, Florida, and contains 29 features describing the demographics and criminal history of the defendants. The binary target label is positive if the defendant did not re-offend within two years after the arrest (55% of the data) and negative if they did (45% of the data). Numerical features are standardized and categorical variables are one-hot-encoded. The one-hot-encoded features are then perturbed in the same fashion as the numerical features and then rounded to the closest binary value for the final residuals.[7] We randomly assign 80% of samples to the train set and the remaining 20% to the test set. A neural network with three hidden layers is trained and reaches an accuracy of 69.72% on the test set. A threshold of 0.5 is chosen for determining whether an individual will recidivate ($<0.5$) or not ($\geq 0.5$).

Table 3 presents the results for the counterfactual search methods on the COMPAS test set. Similar to previous experiments, the RGD approach tends to produce unrealistic counterfactuals with large increases to the classifier's prediction. Conversely, CSGP typically leads to small increases of the classifier score but outputs sparser and more realistic perturbations. The regular GAN method achieves decent gains in prediction score and realism but suffers greatly with respect to sparsity and hence actionability. The CounteRGAN methods proposed in this work are more satisfying than RGD in terms of realism and sparsity. They also achieve similar increases of the classifier prediction as CSGP and produce counterfactuals >1,000x faster than RGD and CSGP.

Specific examples are relevant for investigating what, if any, biases a classifier has learned. Table 4 presents one such data point from the test set. Each method is able to generate a counterfactual that successfully reverts the prediction although they propose very different perturbations to the features. Interestingly, the counterfactuals produced by the GAN and CounteRGAN methods for black-box classifiers find that changing the race to "Caucasian" instead of "Black" contributes to reversing the prediction. In addition, the GAN counterfactual also suggests changing the gender from "Male" to "Female". These insights signal that the recidivism predictor likely holds unfair biases. By extension,

---

[7]An alternative approach would be to handle categorical features using pairwise distance measures and multi-dimensional scaling [Van Looveren and Klaise, 2019].

| | White-box classifier | | | | Black-box classifier | | |
|---|---|---|---|---|---|---|---|
| | RGD | CSGP | GAN | CounteRGAN | RGD | CSGP | CounteRGAN |
| ↑ Prediction gain | **0.38 ± 0.01** | 0.06 ± 0.01 | 0.29 ± 0.01 | 0.07 ± 0.01 | **0.38 ± 0.01** | 0.06 ± 0.01 | 0.12 ± 0.01 |
| ↓ Realism | 1.60 ± 0.08 | 0.78 ± 0.06 | **0.57 ± 0.00** | 0.85 ± 0.09 | 1.60 ± 0.08 | **0.77 ± 0.06** | 0.93 ± 0.09 |
| ↓ Sparsity | 2.07 ± 0.05 | **0.53 ± 0.08** | 7.32 ± 0.16 | 0.85 ± 0.05 | 2.07 ± 0.05 | **0.50 ± 0.08** | 1.48 ± 0.08 |
| ↓ Latency (ms) | 1,704.62 ± 2.12 | 3,312.14 ± 5.46 | **1.39 ± 0.01** | 1.43 ± 0.01 | 3,005.13 ± 2.35 | 9,894.08 ± 51.75 | **1.42 ± 0.12** |
| ↓ Batch latency (s) | 2,459.76 | 4,779.42 | **2.00** | 2.06 | 4,336.40 | 14,277.15 | **2.04** |

Table 3: COMPAS test data results (mean and 95% confidence interval). The arrows indicate whether larger ↑ or lower ↓ values are better, and the best results are in bold. The realism metric typically ranges from 0.87 (mean reconstruction error on the test set) to 5.43 (reconstruction error of random uniform noise in $[0, 1]$).

| | Initial values | White-box classifier | | | | Black-box classifier | | |
|---|---|---|---|---|---|---|---|---|
| | | RGD | CSGP | GAN | CounteRGAN | RGD | CSGP | CounteRGAN |
| age | 24 | - | +1 | +23 | +6 | - | +2 | +12 |
| priors_count | 3 | -9 | -1 | -4 | -2 | -9 | -1 | -1 |
| days_b_screening_arrest | -1 | -1 | - | -3 | - | -1 | - | -12 |
| sex_Male | 1 | - | - | -1 | - | - | - | - |
| c_charge_degree_M | 0 | - | - | +1 | - | - | - | - |
| c_charge_desc_Pos Cannabis W/Intent Sel/Del | 1 | - | - | -1 | - | - | - | -1 |
| c_charge_desc_Possession of Cocaine | 0 | - | - | - | - | - | - | +1 |
| race_Caucasian | 0 | - | - | +1 | - | - | - | +1 |
| Classifier Prediction (prob of not recidivating) | 0.36 | 0.99 | 0.50 | 0.87 | 0.71 | 0.99 | 0.52 | 0.54 |

Table 4: Comparison of counterfactual examples produced by different methods given a sample data point from the COMPAS recidivism dataset. Some of the counterfactuals suggest changing the race and gender indicating potentially unfair biases.

these biases can also be manifest in the COMPAS dataset. This is not necessarily certain, however, since it may have been by chance that the training subset was unbalanced and the model simply picked up on these spurious biases. Though general conclusions should be based on subsequent analysis of complete datasets, counterfactuals such as these can help to probe a classifier's decision boundary in the vicinity of individual data points. Insights such as these illustrate the potential counterfactuals have in helping to audit the fairness of ML systems which should be of paramount relevance to all practitioners.

# 5 DISCUSSION

The proposed CounteRGAN approach applies a novel Residual GAN (RGAN) together with a fixed target classifier to produce realistic and actionable counterfactuals that achieve favorable prediction increases at low fixed latencies. CounteRGAN's separate value functions allow it to be effective even when the target classifier is non-differentiable or a black-box model. In experiments on two diverse datasets, the CounteRGAN produces counterfactuals between 2 and 7 orders of magnitude faster than two state-of-the-art methods. The drop from seconds to milliseconds opens up the possibility of real-time applications. Overall, the resulting counterfactuals are more realistic than competing methods and generally match or exceed prediction gain and actionability. This approach has also shown promise for probing a

classifier's decision boundaries and highlighting potentially unfair biases in use cases such as criminal justice that can have significant stakes for users. Meaningful counterfactuals, such as those produced using the CounteRGAN method, can provide real-time recourse to users and help improve model interpretability and fairness. Together, these form the critical foundations for building effective, scalable, and trustworthy ML systems.

Several promising areas outside the scope of this work are left for future research. These include investigating additional techniques to quantify and ensure actionability, addressing partially mutable features, applying the RGAN and CounteRGAN to additional domains, and experimenting with iteratively improving the counterfactuals by creating a feedback loop to the generator.

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
