# OpenReview forum: "CounteRGAN: Generating Counterfactuals for Real-Time Recourse and Interpretability using Residual GANs"
_auai.org/UAI/2022/Conference — UAI 2022 Poster_

### Official Review · Reviewer_4BUC · 2022-04-10

**Q2(1) Originality/Novelty:** 2
**Q2(2) Significance/Impact:** 2
**Q2(3) Correctness/Technical Quality:** 3
**Q2(6) Clarity Of Writing:** 3
**Q6 Overall Score:** 5
**Q8 Confidence In Your Score:** 2

**Q1 Summary And Contributions:**

The authors introduce RGAN, a residual GAN which leads to counteRGAN, a GAN based method for the computation of counterfactuals for explanations. Furthermore the introduce counteRGAN bb, a method aiming to compute counterfactual explanations without access to the models derivative's. They then compare against other counterfactual explanation methods on MNIST and COMPAS.

**Q2 Assessment Of The Paper:**

More detailed information regarding each of these aspects is given below:

**Q2(4) Quality Of Experiments (Optional):**

3: Good: The experimental evaluation is adequate, and the results convincingly support the main claims.

**Q2(5) Reproducibility:**

3: Good: Key resources (e.g., proofs, code, data) are available and key details (e.g., proofs, experimental setup) are sufficiently well-described for competent researchers to confidently reproduce the main results.

**Q3 Main Strengths:**

The counteRGAN method is a good way to leverage one of the major deep learning successes of GANs to counterfactual explanations. The results generated on MNIST certainly look more realistic than any of the other methods. Furthermore the method is clear and easy to follow.

**Q4 Main Weakness:**

My main concern is that by fitting a GAN to generate counterfactuals it feels like you are passing the interpretability problem one level up. The explanation will now consist of saying this is the closest counterfactual for the individual but when asked why this is the closest counterfactual the answer would then be because an unexplainable deep learning model says so. That is it feels like it could be challenging to interpret such explanations. As the authors comment in the COMPAS example they seem to find unfair bias however there are other explanations such as correlations in the training data which would not be unfair that could account for these. The fact it is hard to see what the correct way to interpret such counterfactuals seems a drawback of this method. Furthermore as the counteRGAN is trained on training data this problem would be increased when the predictor and its explainer were deployed.

**Q5 Detailed Comments To The Authors:**

- The authors mentioned that they do not compare with other methods mentioned in the related work section as they do not address realism, however it seems to me that Wachter et al also doesn't address realism only the nearest prediction. This can be seen in the MNIST example where Wachter et al only changes a few pixels and produces an unrealistic seeming sample.
- As mentioned in the weaknesses section I am unclear if this just passes up the interpretability issue. To say a bit more on this, it seems this wouldn't be a criticism of Wachter et al as they have a clear interpretation of their explanation due to the simple formulation of how the counterfactuals are constructed. As in this case we are fitting another black box to construct counterfactuals it could be challenging to understand why specific counterfactuals are given.
- The above problem would be exaggerated when the model and its predictor are deployed as real world data may not be similar to training data. It would feel important that an explanation would be able to still work in an interpretable in this context to see if our model is making decisions on the incorrect features. However this seems like it could be a problem for this method and we aren't able to clearly comprehend why a specific counterfactual is given.

**Q7 Justification For Your Score:**

This appears like an interesting proposal and they the authors show some improvements on the benchmarks they use. My only concern would be using one black box to explain another, however it may be the case this is still standard in the counterfactual explanations literature.

**Q9 Complying With Reviewing Instructions:**

1: Yes.

---

### Official Review · Reviewer_rAZe · 2022-04-10

**Q2(1) Originality/Novelty:** 2
**Q2(2) Significance/Impact:** 3
**Q2(3) Correctness/Technical Quality:** 3
**Q2(6) Clarity Of Writing:** 4
**Q6 Overall Score:** 4
**Q8 Confidence In Your Score:** 4

**Q1 Summary And Contributions:**

This paper proposes to use a Residual Generative Adversarial Network (RGAN) for counterfactual generation in a supervised learning model. The algorithm can run 2-7 orders of magnitude faster than some of the existing methods. Its counterfactual generation performance ranges from comparable to mildly worse compared to existing algorithms.

**Q2 Assessment Of The Paper:**

More detailed information regarding each of these aspects is given below:

**Q2(4) Quality Of Experiments (Optional):**

2: Fair: The experimental evaluation is weak: important baselines are missing, or the results do not adequately support the main claims.

**Q2(5) Reproducibility:**

3: Good: Key resources (e.g., proofs, code, data) are available and key details (e.g., proofs, experimental setup) are sufficiently well-described for competent researchers to confidently reproduce the main results.

**Q3 Main Strengths:**

In the counterfactual generation problem, the proposed algorithm CounterRGAN is much faster than existing methods with comparable results in prediction gain and actionability.

**Q4 Main Weakness:**

The performance of the proposed algorithm in many cases are comparable or mildly worse than the state-of-the-art algorithms published in 2018 and 2019. It is not clear that the computational gain is sufficient to justify the importance of the proposed algorithm.

The proposed idea is a direct application of RGAN.

The main advantage (time complexity) of CounterRGAN comes from the GAN-based structure. The paper lacks comparison with other GAN-based algorithms without mode collapse issues.


**Q5 Detailed Comments To The Authors:**

The performance of the proposed algorithm in many cases are comparable or mildly worse than the state-of-the-art algorithms published in 2018 and 2019. First, are there newer papers in this area from 2019 to now that generate better performance? Second, it is not clear that the computational gain is sufficient to justify the importance of the proposed algorithm. Specifically, an existing algorithm may take a few seconds while the proposed algorithm can take a few ms. To human users, how significant is this difference, especially if the performance degrades?


The authors argue RGAN can alleviate mode collapse compared with GAN.  Why is this important for the counterfactual applications if mode collapsing does not negatively affect the performance evaluation metrics (as seen in evaluation results) ? Could do you provide some intuition or evidence for "GAN lacks realism as well as actionability due to mode collapse"? Could it be caused by other reasons?

The advantage of the time complexity of CounteRGAN compared to RGD and CSGP is from the GAN-based structure. The paper mentions the benefit of using RGAN instead of GAN is to alleviate mode collapse which will cause lack realism and actionability. But there are some other works to alleviate mode collapse of GAN, for example, Veegan[1]. The performance should be compared with these types of algorithms in the counterfactual generation problem (similar formulated as section 3.3). In the counterfactual generation, what's the benefit of using RGAN compared with other GAN-based algorithms that can alleviate mode collapse?

[1] Srivastava, Akash, et al. "Veegan: Reducing mode collapse in GANs using implicit variational learning." Advances in neural information processing systems 30 (2017).

The loss function of RGAN is similar to equations 2, 6, and 7 in [Shen and Liu, 2016]. The network work structures of RGAN used in simulation, CNN and a neural network with three hidden layers, are classical neural network structures. Could you compare RGAN with [Shen and Liu, 2016] regarding novelty?

Additional reference:
Sauer, Axel, and Andreas Geiger. "Counterfactual Generative Networks." International Conference on Learning Representations. 2020.




**Q7 Justification For Your Score:**

The main concerns are novelty (a direct application of RGAN) and significance (in terms of performance) as discussed above.

**Q9 Complying With Reviewing Instructions:**

1: Yes.

---

### Official Review · Reviewer_vfFW · 2022-04-12

**Q2(1) Originality/Novelty:** 2
**Q2(2) Significance/Impact:** 3
**Q2(3) Correctness/Technical Quality:** 4
**Q2(6) Clarity Of Writing:** 4
**Q6 Overall Score:** 7
**Q8 Confidence In Your Score:** 4

**Q1 Summary And Contributions:**

A GAN approach to generate meaningful counterfactuals that can be used to influence the output of a classifier. The focus is on counterfactuals that are realistic, but at the same time that are actionable, i.e., interpretable and reasonable, in order to help the user to achieve the desired prediction outcome. In addition, counterfactual must be computationally efficient to use them in real-time applications (in particular to provide real-time recourse to the users).

**Q2 Assessment Of The Paper:**

More detailed information regarding each of these aspects is given below:

**Q2(4) Quality Of Experiments (Optional):**

3: Good: The experimental evaluation is adequate, and the results convincingly support the main claims.

**Q2(5) Reproducibility:**

1: Poor: Key details (e.g., proof sketches, experimental setup) are incomplete/unclear, or key resources (e.g., proofs, code, data) are unavailable.

**Q3 Main Strengths:**

Possibility to use the approach for real-time applications thanks to the low values of computational latency


**Q4 Main Weakness:**

The use of a limited amount of datasets.


**Q5 Detailed Comments To The Authors:**

- Page 2: “A realistic counterfactual doesn’t necessarily result in actionable change.”  —> “A realistic counterfactual does not necessarily result in actionable change.”


**Q7 Justification For Your Score:**

The paper provide an interesting idea that can be used not only for provide model interpretability, but that can be also used to help the users in changing some behaviors in order to achieve the desired outcome.


**Q9 Complying With Reviewing Instructions:**

1: Yes.

---

### Official Review · Reviewer_sUGN · 2022-04-14

**Q2(1) Originality/Novelty:** 2
**Q2(2) Significance/Impact:** 2
**Q2(3) Correctness/Technical Quality:** 2
**Q2(6) Clarity Of Writing:** 3
**Q6 Overall Score:** 6
**Q8 Confidence In Your Score:** 3

**Q1 Summary And Contributions:**

This work presents a Residual Generative Adversarial Network-based approach for generating counterfactuals. The proposed approach achieves strong performance in various experiments and could provide insights into model interpretability and fairness issues.

**Q2 Assessment Of The Paper:**

More detailed information regarding each of these aspects is given below:

**Q2(4) Quality Of Experiments (Optional):**

3: Good: The experimental evaluation is adequate, and the results convincingly support the main claims.

**Q2(5) Reproducibility:**

2: Fair: Key resources (e.g., proofs, code, data) are unavailable but key details (e.g., proof sketches, experimental setup) are sufficiently well-described for an expert to confidently reproduce the main results.

**Q3 Main Strengths:**

The experiments are relatively detailed. The proposed approach achieves strong empirical results.

**Q4 Main Weakness:**

1. Although the proposed approach demonstrates effectiveness empirically, its theoretical insights are not discussed sufficiently.

2.  It is unclear whether the evaluation metrics in Table 1, e.g., the formula of "realism" is a reasonable metric.

3. The authors only considered low-resolution MNIST for image datasets. It would be more convincing to also consider more challenging higher-resolution datasets.

**Q5 Detailed Comments To The Authors:**

1. It would be good to provide more theoretical insights into why the proposed approach works.
2. It is unclear whether the metric "realism", defined in Table 1 is a reasonable metric. It seems that if the autoencoder is not well trained, the metric could be off. It is also unclear whether L2 distance is the most reasonable distance to compare in all settings.
3. The image dataset experiment is only on MNIST, which is a low-resolution dataset. It would be more convincing to also consider other more challenging higher-resolution datasets, like CIFAR-10, CelebA.



**Q7 Justification For Your Score:**

This work proposes an approach to generate meaningful counterfactuals for a target classifier. Although the reported experimental results are good, it is unclear whether certain evaluation metrics are defined properly. At the same time, theoretical insights of the proposed approach are not properly discussed. The image dataset considered is also small and low resolution, it would be more convincing to consider more challenging datasets like CIFAR-10, CelebA.

**Q9 Complying With Reviewing Instructions:**

1: Yes.

---

### Decision · Program_Chairs · 2022-05-15

**Decision:**

Accept (Poster)

**Comment:**

Meta Review: Thank you for your submission to UAI and your author response to reviews.

This paper presents a method for using a Residual Generative Adversarial Network for counterfactual generation, subject to the counterfactual example giving a prediction gain, adhering to realism, and to actionability.  The counterfactual provides insight into the explainability of the model and, with correct formulation of actionability of perturbations that can transform an input into its counterfactual, provides recourse information as well.  Reviewers appreciated the importance of this task, the empirical evaluation, and speed of generation.

The paper could be improved through a stronger statement of theoretical insights; larger datasets in empirical results; comparison to (more) baseline methods; and a discussion of the challenges of identifying a method for determining actionability (and realism) that can constrain counterfactual generation.